# Ability of BlueCheck Liquid for Detection of Initial Lesions on Occlusal Surfaces In Vitro

**DOI:** 10.3390/diagnostics15243145

**Published:** 2025-12-10

**Authors:** Anahita Jablonski-Momeni, Mara Müller, Peter Bottenberg

**Affiliations:** 1Department of Orthodontics, Medical Faculty, Dental School, Philipps-University Marburg, 35039 Marburg, Germany; 2Department of Oral Health Care, Université Libre de Bruxelles (ULB), 1050 Brussels, Belgium; peter.bottenberg@ulb.be; 3 University Center for Dental Care, Vrije Universiteit Brussel, Laarbeeklaan 103, 1090 Brussels, Belgium

**Keywords:** caries detection, activity assessment, occlusal surfaces, hemoglobin-based liquid, ICCMS, histology

## Abstract

**Background/Objectives:** Visual examination remains the standard for caries detection, but its subjectivity limits reproducibility. Adjunctive methods may support objectivity, yet many require costly equipment or extensive training. BlueCheck (BC) liquid is a hemoglobin-based dye that binds reversibly to porous hydroxyapatite, producing a visible blue discoloration in demineralized enamel. This study aimed to evaluate the ability of BC liquid to detect initial occlusal caries lesions and assess lesion activity. **Methods:** An in vitro study was performed on 54 extracted permanent posterior teeth. Caries detection and lesion activity assessment were performed visually (International Caries Classification and Management System) and with BC liquid. Histology, including methyl red staining, served as the reference standard for lesion depth and activity. Agreement, sensitivity, specificity, and ROC analyses were calculated. **Results:** BC liquid showed almost perfect intra- and inter-examiner reproducibility (κ = 0.86). Overall agreement with visual examination was 92.6%, and agreement with histological activity assessment was 79.6%. For detection of initial lesions, BC liquid achieved sensitivity of 81.8% and specificity of 100%, with an AUC of 0.909. For activity assessment, BC liquid showed higher sensitivity than visual examination (89.5% vs. 60.5%), but lower specificity (75% vs. 100%), yielding an AUC of 0.822. **Conclusions:** Within the limitations of this in vitro study, BC liquid demonstrated good diagnostic accuracy and agreement with visual criteria for both detection and activity assessment of occlusal caries lesions. BC liquid may serve as an adjunctive tool for caries diagnosis, but further in vivo and longitudinal studies are required to validate its clinical applicability.

## 1. Introduction

Clinical visual detection of caries lesions is recommended as the first-choice method for the detection and assessment of caries lesions on accessible surfaces [1]. However, visual caries detection is often subjective [2] and adjunct methods can provide additional benefit in objectifying and documenting demineralization. Detecting initial-stage caries lesions may be more difficult as they develop in areas of plaque stagnation [3]; thus, removing plaque is essential before clinical examination.

Various methods for the operator-independent detection of initial caries lesions have been evaluated in the past [4,5,6,7,8]. However, many appliance-based methods are rarely adopted in daily clinical practice because of their cost and the training they require. An objective and reproducible method for supporting caries detection is more likely to be integrated into routine procedures if additional expenses and complexity are kept to a minimum. Many available procedures and appliances such as optical methods (fluorescence-based cameras or devices) are connected with additional hard- and software acquisition, need technical know-how, and may be simply too onerous for practitioners [9].

A novel diagnostic method named BlueCheck was introduced for detection of initial carious lesions via reversible staining of porous bioapatite (Incisive Technologies Pty Ltd. Melbourne, Australia). It utilizes the natural hydroxyapatite-binding chemistry of proteins to specifically and reversibly bind to porous dental hydroxyapatite [8]. The BlueCheck indicator (hereinafter referred to as BC liquid) consists of a deep-blue dye (Amido black) linked to a protein (hemoglobin) with high affinity for porous hydroxyapatite (European Patent No. 2 547 311 B1). This targeted binding to demineralized enamel enables high-contrast visualization of carious lesions directly on the tooth surface, without the need for additional equipment. The interaction is electrostatic and reversible; the blue color can be removed easily by brushing with standard toothpaste containing a detergent.

A proof-of-concept study already showed the ability of the BC liquid for detection of artificial demineralization on smooth surfaces in vitro [9]. Furthermore, it was shown that the BC liquid was sufficiently sensitive to distinguish between unaffected enamel and artificial caries-like lesions [10]. Moreover, the method showed good accuracy in visualization of demineralized dentin in vitro [11,12].

To date, no published studies have investigated the use of this tool on natural enamel caries lesions. Therefore, the present study aimed to assess the ability of BlueCheck to stain initial occlusal caries lesions and to differentiate between active and inactive lesions in vitro using histology as reference.

## 2. Materials and Methods

### 2.1. Tooth Selection and Sample Size Calculation

The study was performed on human extracted permanent posterior teeth. The use of extracted teeth was approved by the Ethics Committee of the Medical Faculty of the Philipps-University of Marburg, Germany (approval number (AZ 132/19). Prior to extraction written informed consent was obtained from each patient for the use of the extracted teeth for study purposes. The research did not involve human participants. All methods were carried out in accordance with relevant guidelines and regulations.

After extraction, teeth were stored in a 0.9% NaCl solution containing 0.001% sodium azide at 6 °C. Prior to the cleaning procedure, each surface was examined for the presence or absence of thick plaque on the occlusal surface. Teeth were then cleaned with scalers and a rotating bristle (Pluradent GmbH, Offenbach, Germany) and polishing paste (Clinpro Prophy Paste, 3M ESPE, Seefeld, Germany). The remaining paste was removed with a multifunctional syringe using water and air, and the teeth were stored in deionized water. The subsequent assessments took place within 10 days following tooth extraction.

The final selection of the teeth was performed after air drying and observation under a microscope at 10× magnification (Leica Z6 APO, Wetzlar, Germany). Teeth with developmental defects, fluorosis, and other opacities or erosive wear were excluded.

Sample size calculation was performed with MedCalc for Windows, version 20.010 (MedCalc Software, Ostend, Belgium, www.medcalc.org). Based on pilot examinations (unpublished data), a correlation of 0.7 was assumed between BC liquid and visual findings for lesion activity assessment. A sample size of 17 teeth was calculated for a power of 90% and α = 0.05. Allowing for the potential loss of one tooth during sample handling, 18 teeth were allocated to each group (sound, active, and inactive lesions).

### 2.2. Visual Examination

One site within the pit and fissure system of each tooth was selected by two examiners experienced in cariology, both previously involved in studies using the International Caries Classification and Management System (ICCMS).

Lesion extent and activity were assessed using consensus scores. Each tooth was embedded in a silicone block (Silaplast Futur, Detax GmbH, Ettlingen, Germany), and digital images of the occlusal surfaces were obtained. Investigation sites were defined by the examiners on the tooth surface and subsequently marked on the corresponding digital images. For site localization during the study, draft-quality copies of the images were printed on plain paper and used by the examiners.

The occlusal surfaces were classified by consensus using the ICCMS™ Caries Merged categories [13]:

Sound surfaces (Code 0): no evidence of visible caries (no or questionable change in enamel translucency) when viewed clean and after prolonged air-drying (5 s).

Initial-stage caries (codes 1 and 2): First or distinct visual changes in enamel seen as a carious opacity or visible discoloration (white spot lesion and/or brown carious discoloration) not consistent with clinical appearance of sound enamel.

Lesion activity assessment involved the following characteristics for active lesions [13]: the surface of the enamel is whitish/yellowish, is opaque with a loss of luster, and feels rough when the tip of the ball-ended probe is moved gently across the surface. The lesion is in a plaque stagnation area (e.g., in the entrance of pits and fissures) and may be associated with large amounts of accumulated biofilm. Inactive lesions were classified when the surface of the enamel was whitish, brownish, or black. Enamel is shiny and feels hard and smooth when the tip of the ball-ended probe is moved gently across the surface. The lesion surface is free of visible biofilm accumulation.

### 2.3. Measurements with BlueCheck (BC Liquid)

As baseline measurements all samples were documented photographically using a dental camera (dentaleyepad, doctorseyes GmbH, Ochsenhausen, Germany) and a microscope at 10× magnification (Leica Z6 APO and QWin Standard V 3.4.0 Software, Leica Microsystems, Wetzlar, Germany). Then the BC liquid was applied using a microbrush and left undisturbed for 3 min according to manufacturer’s instructions, followed by a rinse with water and air drying the surface. Each surface was examined for the presence of blue areas by two examiners, first by naked eye and then under 10× magnification, independently and consecutively. In addition, standardized photographs were taken to document the findings.

Afterwards the samples were cleaned in an ultrasound bath containing 10% sodium lauryl sulphate (SLS) as to manufacturer’s instructions. A second BC liquid application was performed to evaluate reproducibility of the findings by each examiner. Staining of each occlusal surface was documented dichotomously (yes/no), with no additional grading or categorization applied.

### 2.4. Histological Sections

The histological methodology was performed following a standardized protocol. The crown of each tooth was hemisectioned (200-μm-thick sections) at the investigation site with a saw band with grade D64 diamonds (Exakt, Hamburg, Germany). The following histological classification was used to record caries severity at each investigation site [14]: Score 0: No enamel demineralization or a narrow surface zone of opacity (edge phenomenon); Score 1: Enamel demineralization limited to the outer 50% of the enamel layer; Score 2: Demineralization involving the inner 50% of the enamel, up to the enamel-dentine junction; Score 3: Demineralization involving the outer 50% of the dentine.

Lesion activity was determined by applying a 0.1% methyl red solution to each section, which served as a pH indicator [15,16]. Lesions with pH < 5.5 stained red/pink (active), whereas those with pH > 5.5 appeared yellow (inactive). After removal of excess dye with a cotton bud, sections were examined under 2.5× magnification. Both lesion depth and histological activity assessment were performed based on a consensus decision of the same examiner who carried out the BC liquid measurements.

Figure 1 and Figure 2 present standard visible images together with the corresponding BC liquid, and histological images, illustrating a sound surface and a surface with an initial active lesion. In Figure 1b, no blue coloration is visible on the enamel surface after rinsing off the BC liquid. A slight bluish area outside the fissure system can be observed, confirming that the agent had been applied to the sample. Importantly, no staining persisted within the fissure itself. In Figure 2b, by contrast, the fissure shows clear blue staining after application of the BC liquid and subsequent rinsing, indicating the presence of a carious lesion. The histological section after application of methyl red dye shows a distinct red staining (Figure 1d), corresponding to an active carious lesion.

### 2.5. Statistical Evaluation

Statistical analysis was performed using MedCalc Statistical Software version 23.3.7 (MedCalc Software Ltd., Ostend, Belgium; https://www.medcalc.org; 2022).

Cohen’s kappa was used to calculate examiner agreement for BC liquid assessments. McNemar test was used to test whether there were significant differences between the measurements. Correlation between all methods was calculated using Spearman’s rank correlation coefficient (rs). The results from the histology were used to calculate sensitivity (SE) and specificity (SP). For activity assessment, histological scores (activity +/−) were used for calculation of SE and SP. Using sensitivity and specificity values, ROC analyses were carried out. The areas under the ROC curves of visual and bioluminescence methods were compared [17]. The level of significance was set at α = 0.05.

## 3. Results

For the in vitro study, 54 unrestored permanent posterior teeth were available. Of these, 12 were classified as sound and inactive by histology, 42 showing various grades of carious lesions of which 38 were classified as active.

Intra-examiner agreement for the BC liquid readings was 100% for examiner 1 and 2. Inter-examiner agreement was κ = 0.86 (95% CI 0.706; 1.00), corresponding to an almost perfect agreement.

Percentage agreement between visual and BC liquid measurements was 92.6%, irrespective of activity assessment. Agreement between activity assessment and BC liquid staining was 79.6%, showing 4 false negatives and 4 false positives for BC staining. The distribution of ICCMS cross-tabulated with the BC measurement results is presented in Table 1.

Significant correlation (*p* < 0.0001) was found between all methods. Spearman correlation coefficients ranged between −0.805 (ΔF vs. histology depth) and 0.832 (visual caries classification and histology depth). All results are displayed in Table 2.

In Table 3 and Table 4, the distribution of the histology scores with regard to lesion depth and activity are shown, cross-tabulated with the ICCMS and BC liquid findings. Histologically, 29.6% of occlusal surfaces were classified as sound or inactive in the stained dyed sections and 70.4% showed staining in the sections, acting as a sign for active lesions (Table 4). Compared to this reference standard, 89.5% (n 34 out of 38) of the active lesions stained positively with BC, while the percentage agreement for inactive lesions amounted to 75% (n 12 out of 16).

Figure 3 shows the ROC curves for visual and BC liquid assessment using histological depth as a reference standard, while Figure 4 shows the ROC curves using histological activity as reference standard.

Table 5 shows the corresponding areas under the curves, sensitivity, and specificity values for visual and BC liquid findings. The AUC of the methods showed comparable results, without statistical differences. However, Sn and Se values varied depending on the method. For caries detection of initial lesions, highest sensitivity value was obtained by visual detection (100%), followed by BC liquid (81.8%). The highest sensitivity value for assessment of lesion activity was obtained by BC liquid (89.5%), followed by visual assessment (60.5%).

## 4. Discussion

BC liquid as a newly developed diagnostic method has so far been investigated in only a limited number of studies. The present work is the first to evaluate its use for two distinct diagnostic purposes: detection of initial occlusal caries lesions and assessment of lesion activity. Caries diagnosis is still too often restricted to determining the presence or absence of cavitation. In contrast, early detection enables clinicians to arrest disease progression or manage lesions with non-operative or minimally invasive approaches. Therefore, effective caries management requires diagnostic methods and indices that capture the full spectrum of lesion severity as well as their activity status [1]. A wide range of methods for detecting caries on different tooth surfaces have been developed and evaluated. In daily practice, however, only visual examination, visual-tactile examination, and radiography have achieved global clinical application [1]. Adjunct methods, such as light-based or dye techniques, are not routinely used in daily practice worldwide due to cost, practicality, or reimbursement issues. Nevertheless, they remain of considerable scientific interest.

Beyond availability and ease of use, the clinical value of any diagnostic method depends on its validity—with accuracy and reproducibility being key factors. From the practitioner’s perspective, adjunct methods should be applicable to all tooth surfaces in both dentitions, demonstrate high diagnostic accuracy and reproducibility (particularly for proximal and occlusal sites), be easy to operate, and ideally provide imaging features that objectively document changes over time in relation to relevant anatomical structures (e.g., dental pulp), rather than offering only categorical or numeric scores [1]. Yet, no adjunct method has fully met this standard. Current methods for diagnosing caries still show certain limitations. Therefore, radiation-free diagnostic tools with reduced operator dependency are needed. Such approaches may increase patient acceptance by making caries detection more transparent and encouraging adherence to preventive and minimally invasive care. This underscores the need for researchers and manufacturers to further improve existing devices or develop innovative methods for caries detection.

Our findings indicate that BC liquid shows high agreement with visual methods for lesion detection and activity assessment in vitro (Table 1), and its diagnostic accuracy was comparable to visual examination. BlueCheck acts by binding directly to mineral crystals exposed through demineralization, producing a visible blue discoloration at affected sites. This provides an immediate clinical readout of lesion location without the need for additional scoring systems and may facilitate communication with patients. However, no validated scale currently exists to distinguish or classify lesion severity based on the intensity of blue staining, which limits the method’s potential for quantification. In an in vitro study, Lippert et al. [8] investigated BlueCheck for the detection of artificial enamel caries-like lesions of varying severities, using spectrophotometry to measure color changes. Similar to our findings with extracted human teeth, they observed no discoloration in sound enamel, whereas all demineralized specimens stained blue with increasing intensity correlating positively with demineralization time [10]. Preliminary, unpublished findings indicate that BC liquid may help visualize areas of ongoing remineralization on tooth surfaces. Likewise, initial observations suggest that BlueCheck, despite containing a hydroxyapatite-binding protein, does not impair the remineralization of artificial enamel caries. These early insights are linked to the question of whether blue staining can be quantified, an aspect that requires further investigation. Future research including teeth with wider variability in lesion depth could enable a more detailed evaluation of staining intensity and help determine its potential clinical relevance.

The findings of the present study are consistent with the previous results on the use of BC liquid in artificially induced enamel lesions on smooth surfaces, both with and without orthodontic brackets [9]. In that study, the liquid successfully stained the artificial lesions, with sensitivity and specificity values reported as 100%. These results were superior to those of the present investigation, which may be explained by the use of natural lesions in our study, as they are inherently more heterogeneous than artificially produced ones.

The present findings also highlight the potential clinical usefulness of BlueCheck in everyday practice, particularly in orthodontic and pediatric settings. Both patient groups are at increased risk for the development of early enamel demineralization, and a simple, chairside method that enhances the visibility of incipient lesions could support timely preventive interventions. In orthodontic patients, BlueCheck may aid in detecting early changes around brackets, where conventional visual inspection is often challenging. In pediatric dentistry, the method could assist in identifying early lesions in uncooperative or anxious children by providing an immediate and easily interpretable visual cue. By improving the identification of early-stage lesions, BlueCheck may help clinicians tailor preventive strategies more effectively and reinforce oral-hygiene instructions based on visible findings. This again underscores the need for clinical studies in different patient populations to confirm the potential benefits of BlueCheck.

Regarding detection of initial occlusal caries lesions, the performance of BC liquid was comparable to, or better than, that of modern adjuncts reported in recent studies. For example, targeted fluorescent starch nanoparticles (LumiCare™ Caries Detection Rinse, GreenMark Biomedical Inc.) achieved substantial to high inter-user agreement, with sensitivity and specificity of 88.9% and 94.6%, respectively, for early occlusal lesion detection [18]. In comparison, BC liquid demonstrated similar diagnostic accuracy, highlighting its potential as a competitive adjunct for the detection of early lesions. A recent study directly compared the two caries detection methods, BlueCheck and LumiCare, on smooth surfaces of extracted human teeth [11]. The results showed that both methods were substantially equivalent and performed favorably compared to traditional visual detection. A recent review highlighted that currently available digital diagnostic aids for caries detection—including radiation-, light-, ultrasound-, and electric-based methods—offer certain advantages but also notable limitations. While they can support clinicians as adjuncts to conventional caries detection, none have demonstrated ideal performance, particularly in identifying early-stage lesions [19].

One in vitro study has examined the use of BlueCheck for detecting dentin caries [12]. In that work, Caries Finder and BlueCheck were compared to assess whether they were substantially equivalent in visualizing demineralized dentin and to evaluate their performance relative to traditional visual/tactile detection. The results indicated that both adjunctive methods compared favorably with the conventional approach and showed comparable performance in differentiating carious dentin from sound tooth structure in vitro.

While caries is a dynamic process that can, over time, progress, arrest, or regress, lesion activity is independent of the lesion depth and rather depends on the patient’s ability to remove surface biofilm [3,20]. From a clinical perspective, it is well-established that caries risk/susceptibility is closely linked with caries activity [3]: lesions in high risk patients are more likely to be active and progress than lesions in low-risk patients. Differentiating between active and inactive early lesions is essential for guiding preventive decisions or operative care and for monitoring a lesion. Normally, lesion activity assessment is a clinical task, as activity can only be determined over time [3]. A key point is that methods which assess activity often rely on indirect clinical indicators—porosity, demineralization signs, staining, and changes over time—rather than direct measurement of bacterial activity or mineral content change. This introduces both opportunities (simplicity, practicability) and pitfalls (lower specificity, variability, dependence on sample condition). However, for research purposes, attempts have also been made to evaluate activity in vitro. The histology method used in the present investigation was based on already published methods [15,16,21]. Our results suggest BC liquid-stained active lesion with a proportionally higher rate but that some false positives were recorded in inactive lesions, resulting in some inactive lesions being misclassified as active. For initial enamel lesions, this may simply lead to the application of additional preventive measures. For moderate or severe lesions, however, the risk of overtreatment must be weighed and lesion activity carefully considered using all clinically available signs. Enhanced visualization through BlueCheck may support this process in identifying lesions that require immediate intervention versus those that can be monitored. While the present in vitro findings cannot fully reflect clinical behavior, the concept may help clinicians prioritize non-invasive interventions for active lesions and avoid overtreatment in inactive ones. To clarify the practical value of the method and its role in guiding therapy, well-designed prospective clinical studies are still required. As with any detection device, BC liquid should be regarded as an adjunct to, rather than a replacement for, visual assessment. Another in vitro study using the LumiCare rinse for caries activity assessment reported perfect agreement with visual assessment in distinguishing active from inactive and/or sound surfaces, suggesting potential as an objective indicator of caries activity [22]. It should be noted, however, that in that study the reference standard was visual assessment according to ICCMS, and no histological validation was performed, unlike in the present investigation.

A key limitation of this study lies in its in vitro design. Extracted teeth provide a controlled and standardized setting, but they cannot fully reflect the dynamic and multifactorial conditions of the oral cavity. In vitro samples lack physiological factors such as continuous salivary flow, pellicle renewal, temperature fluctuations, and mechanical influences from mastication or oral hygiene procedures. Moreover, the absence of natural biofilm maturation and surface moisture may alter how the staining agent interacts with enamel. These constraints can influence stain uptake, distribution, and visual contrast when compared to in vivo examinations. Additionally, clinical factors such as patient movement, soft-tissue interference, and variability in ambient lighting conditions may further affect the method’s diagnostic performance. Therefore, the results should be interpreted with caution, and well-designed clinical studies are required to validate the reproducibility and clinical applicability of the method. Regulatory aspects also played a role. The batch of BC liquid used in this study was approved for laboratory use only. Since completion of the study, however, the product has obtained the necessary approval for clinical application in the United States, ensuring that future in vivo studies can be conducted with the clinically approved formulation (https://www.incisive-technologies.com/product, accessed on 20 October 2025).

Regarding reference standards, as in many activity assessment studies, the gold standard for activity is imperfect. Different studies use different definitions, criteria, or protocols, which affects comparability [23]. In our study, microscopic criteria were used; however, clinical activity indicators such as texture or plaque retention may differ. Moreover, methyl red provides only a chemical proxy for acidity at time of examination rather than a true clinical gold standard, and the resulting sensitivity and specificity estimates should therefore be interpreted with caution.

Considering the limitations of existing adjunctive methods, BC liquid may provide some practical advantages, such as simple application, a quick visual readout without additional equipment, and good diagnostic accuracy. It could be of particular value in settings where advanced imaging tools or AI support is not available. The ability to visualize lesion activity may also facilitate communication with patients and encourage preventive behaviors, for example, in individuals at high caries risk, such as those with fixed orthodontic appliances [9]. Integration with established diagnostic protocols, including visual/tactile criteria and risk assessment, may further improve clinical decision-making. Within the limits of this in vitro study, BC liquid may therefore be considered a supportive adjunct for caries detection and activity assessment, with its clinical usefulness requiring confirmation in future studies.

## 5. Conclusions

Within the limitations of this in vitro study, BC liquid showed good agreement with visual and histological criteria in detecting lesion severity and activity in enamel caries. It was accurate in distinguishing sound from initial occlusal caries lesions, with diagnostic performance comparable to visual examination. These findings suggest that BC liquid may serve as an adjunct to visual caries diagnosis, but confirmation in clinical and longitudinal studies is needed to determine its value for early detection and minimally invasive management.

## Figures and Tables

**Figure 1 diagnostics-15-03145-f001:**
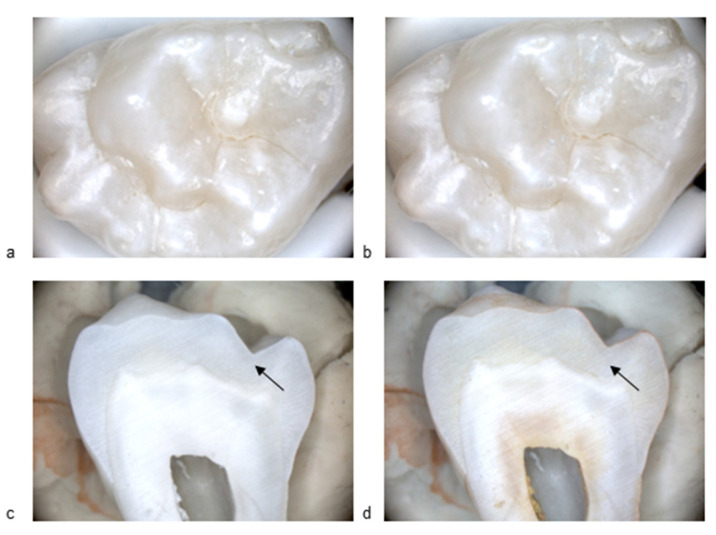
(**a**) Image of the occlusal surface with ICCMS code 0 (caries free, sample #E19), before BC liquid application; (**b**) Corresponding image after BC liquid application; (**c**) Histological section (arrow indicates the region of interest); (**d**) Histological section after application of methyl red. All images at 10× magnification.

**Figure 2 diagnostics-15-03145-f002:**
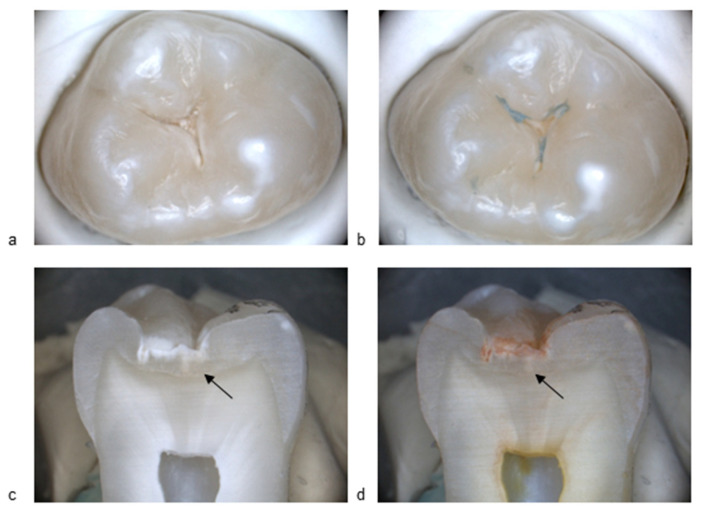
(**a**) Image of the occlusal surface with ICCMS code 1+ (initial active lesion, sample #G17), before BC liquid application; (**b**) Corresponding image after BC liquid application; (**c**) Histological section (arrow indicates the region of interest); (**d**) Histological section after application of methyl red. All images at 10× magnification.

**Figure 3 diagnostics-15-03145-f003:**
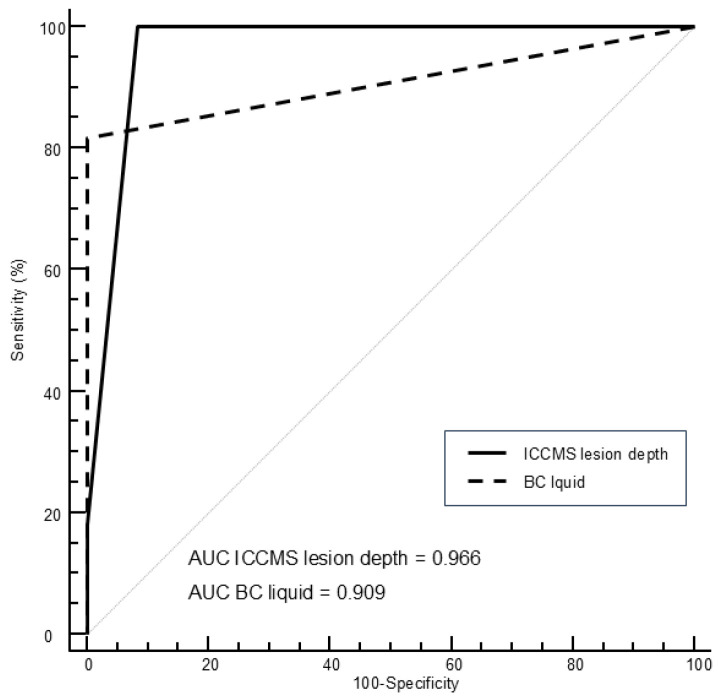
ROC curves for visual and BC liquid assessment using histological depth as a reference standard.

**Figure 4 diagnostics-15-03145-f004:**
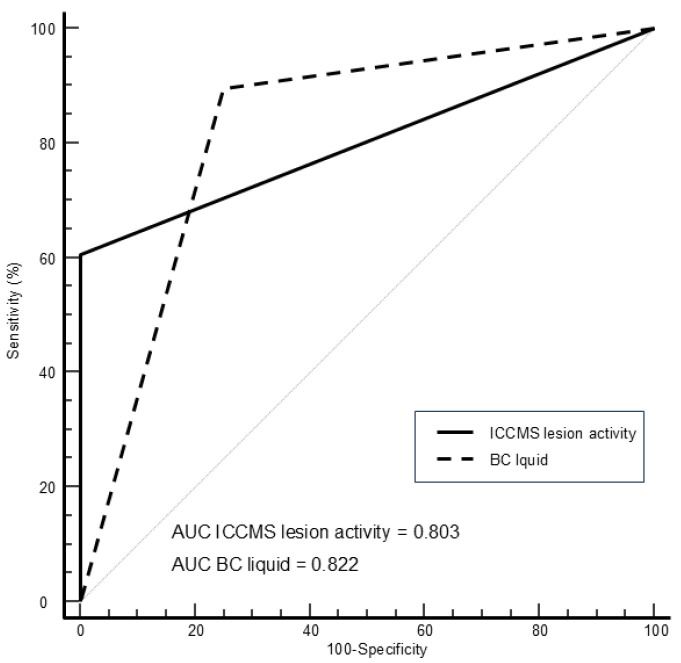
ROC curves for visual and BC liquid assessment using histological activity as a reference standard.

**Table 1 diagnostics-15-03145-t001:** Distribution of ICCMS cross-tabulated with the results of the BC measurements.

	BC Liquid		
ICCMS code	Without staining	With staining	N total (%)
Sound surface (code 0)	12	0	12 (22.2%)
Initial caries lesion (code 1+ and 2+) *	0	27	27 (50%)
Initial caries lesion (code 1− and 2−) *	4	11	15 (27.8%)
N total (%)	16 (29.6%)	38 (70.4%)	54 (100%)

* The suffix + or − indicates an active or inactive lesion.

**Table 2 diagnostics-15-03145-t002:** Spearman correlation coefficients (rs) for all methods (95% confidence interval in parenthesis).

Method	ICCMS Activity	BC Liquid	Histology, Depth	Histology, Activity
ICCMS Caries	0.622 (0.425; 0.763)	0.665 (0.483; 0.792)	0.832 (0.727; 0.900)	0.725 (0.568; 0.831)
ICCMS Activity		0.559 (0.342; 0.719)	0.640 (0.449; 0.775)	0.559 (0.342; 0.719)
BC Liquid			0.737 (0.585; 0.839)	0.645 (0.456; 0.778)
Histology, Depth				0.737 (0.585; 0.839)

**Table 3 diagnostics-15-03145-t003:** Distribution of ICCMS and results of the BC liquid measurements cross-tabulated with the results of the histology for caries depth.

	Histology Scores (Lesion Depth)				
ICCMS code	0	1	2	3	N total (%)
Sound surface (code 0)	12	0	0	0	12 (22.2%)
Initial caries lesion (code 1+ and 2+) *	0	9	14	4	27 (50%)
Initial caries lesion (code 1− and 2−) *	0	13	2	0	15 (27.8%)
N total (%)	12 (22.2%)	22	16	4	54 (100%)
	**Histology scores (Lesion Depth)**				
BC liquid	0	1	2	3	N total (%)
Without discoloration	12	4	0	0	16 (29.6%)
With discoloration	0	18	16	4	38 (70.4%)
N total (%)	12 (22.2%)	22 (40.8%)	16 (29.6%)	4 (7.4%)	54 (100%)

* The suffix + or − indicates an active or inactive lesion.

**Table 4 diagnostics-15-03145-t004:** Distribution of ICCMS and results of the BC liquid measurements cross-tabulated with the results of the histology for caries activity assessment after application of methyl red.

	Histology Scores (Lesion Activity)		
ICCMS code	0 = inactive	1 = active	N total (%)
Sound surface (code 0)	12	0	12 (22.2%)
Initial caries lesion (code 1+ and 2+) *	0	27	27 (50%)
Initial caries lesion (code 1− and 2−) *	4	11	15 (27.8%)
N total (%)	16 (29.6%)	38 (70.4%)	54 (100%)
	**Histology scores (Lesion Activity)**		** **
BC liquid	0 = inactive	1 = active	N total (%)
Without discoloration	12	4	16 (29.6%)
With discoloration	4	34	38 (70.4%)
N total (%)	16 (29.6%)	38 (70.4%)	54 (100%)

* The suffix + or − indicates an active or inactive lesion.

**Table 5 diagnostics-15-03145-t005:** Area under the ROC curve and diagnostic accuracy for all methods.

	Reference Standard: Histological Lesion Depth							
Method	Cut-off	AUC (95% CI)	SE	Sn (%)	Sp (%)	Ppv (%)	Npv (%)	Comparison of AUC
ICCMS caries	Sound vs. initial lesion	0.966 (0.839; 0.999)	0.039	100	91.7	95.7	100	*p* = 0.379
BC liquid	discoloration no vs. discoloration yes	0.909 (0.760; 0.980)	0.052	81.8	100	100	75	
	**Reference standard: Histological Lesion Activity**							
Method	Cut-off	AUC (95% CI)	SE	Sn (%)	Sp (%)	Ppv (%)	Npv (%)	Comparison of AUC
ICCMS activity	Sound surface/inactive lesion vs. active lesion	0.803 (0.672; 0.898)	0.058	60.5	100	100	51.6	*p* = 0.811
BC liquid	discoloration no vs. discoloration yes	0.822 (0.694; 0.913)	0.071	89.5	75	89.5	75.0	

AUC: area under the ROC curve, CI: confidence interval, SE: standard error, Sn: sensitivity, Sp: specificity, Ppv: positive predictive value, Npv: negative predictive value.

## Data Availability

All data generated or analyzed during this study are included in this article. Further inquiries can be directed to the corresponding author. The data sheet will be available in a research data Repository after acceptance of the manuscript.

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
