# Peer review of "Ability of BlueCheck Liquid for Detection of Initial Lesions on Occlusal Surfaces In Vitro"

_diagnostics, 2025, doi:10.3390/diagnostics15243145_

Round 1
Reviewer 1 Report
Comments and Suggestions for Authors
Dear Authors,
The paper is well structured, clearly written and makes a relevant contribution to the diagnosis of incipient caries. The study is rigorous and uses appropriate methods, including histological correlation, which gives it scientific value. The methodology is well described, with correct statistical analyses (ROC, AUC, kappa). The discussions are balanced, and the conclusions reflect the results. The bibliography is updated (2021–2025).
Areas for improvement:
- Lack of clarification regarding the blinding of the examiners – it should be mentioned whether the assessments were independent to avoid bias.
- BC liquid analysis only binary (yes/no) – a quantitative analysis of the staining intensity would be useful.
- The “Limitations” section is too short – it is recommended to complete it with in vitro limitations, the influence of the salivary environment and the need for clinical studies.
- ROC figures (3 and 4) – it is recommended to add AUC values directly to the graph for clarity.
- Clinical applicability – the potential for use in practice (in orthodontic or pediatric patients) should be emphasized.
- Therapeutic orientation – it would be useful for authors to emphasize the clinical implications and therapeutic direction depending on the type of lesion (active/inactive), to highlight the practical value of the method.
Recommendation: Accept with minor revisions.
Author Response
Please see the attachement.

Reviewer 2 Report
Comments and Suggestions for Authors
The work is entirely in vitro. While this is acceptable for preliminary validation, the absence of saliva, pellicle, biofilm, and clinical variability makes it difficult to translate the results into real clinical performance. A second major limitation is the very simplistic dichotomous scoring of BlueCheck staining. Reducing the outcome to “present or absent” overlooks potentially meaningful gradations in staining intensity that could correlate with lesion severity. This limits the diagnostic depth of the method.
Furthermore, the histological assessment of lesion activity using methyl red provides only a chemical proxy for acidity and cannot serve as a true gold standard for clinical activity. Sensitivity and specificity estimates for activity assessment should therefore be interpreted with caution. It would also strengthen the manuscript to include a comparison with at least one other established diagnostic adjunct, as the current comparison is limited to visual ICCMS scoring.
Two additional editorial aspects deserve attention. The manuscript occasionally refers to BlueCheck as a “technique”, but it would be more accurate to describe it as a diagnostic method. There also appears to be a likely figure citation error in the text, where “Figure 1b” seems to refer instead to “Figure 2b”. This should be verified and corrected to avoid confusion.
Overall, the study provides promising preliminary data, but the authors should address these methodological and reporting issues before the manuscript can be considered for publication.
Round 2
Reviewer 2 Report
Comments and Suggestions for Authors
the paper is now ready for the pubblication